# Hybrid Assessment for Strengthening Supply Chain Resilience and Sustainability: A Comprehensive Analysis

El-Awady Attia [1,2,*] and Md Sharif Uddin [1,3]

1   Industrial Engineering Department, College of Engineering, Prince Sattam bin Abdulaziz University,
    Al-Kharj 11942, Saudi Arabia; msharifju@juniv.edu
2   Mechanical Engineering Department, Faculty of Engineering (Shoubra), Benha University, Cairo 11629, Egypt
3   Department of Mathematics, Jahangirnagar University, Savar 1342, Bangladesh
*   Correspondence: e.attia@psau.edu.sa

**Abstract:** Organisations encounter a significant challenge in the globalised business landscape, and thus mitigate risk by establishing robust supply chains (SCs) networks is required. In a rapidly changing environment, gaining a competitive edge is imperative. However, the exploration of the essential factors enabling resilient and sustainable supply chain management (RSSCM) in construction projects has been lacking. This study aims to bridge this gap by identifying the enabling factors for resilient and sustainable supply chain management (SSCM). To achieve this, a survey was conducted among Egyptian engineers, involving 32 factors derived from an extensive literature review on RSSCM. The data collected were categorised into four groups, namely Organisational Knowledge and Competence, Risk Management and Security, Collaboration and Communication, and Planning Efficiency and Timing, using brainstorming techniques. Subsequently, the data were analysed utilising a novel hybrid assessment approach that combines evaluation of alternatives and ranking, employing the compromise solution-fuzzy synthetic evaluation methodology, for the first time, offering a unique approach to assessing and prioritising these categories. The findings reveal that 'Planning Efficiency and Timing' emerged as the highest-performing category, whereas 'Collaboration and Communication' performed the worth. Furthermore, our results indicate that brainstorming enabled the grouping of the enablers into four distinct categories, providing a structured framework for understanding and organising them. The integration of MARCOS and FSE offered a robust decision-making approach, proposing a resilient and comprehensive decision-support system capable of tackling intricate real-world issues. This research outcome offers building administrators valuable insights for comparing different supply chains, considering how supply chain characteristics influence resilience and risk exposure in building SCs.

**Keywords:** importance fuzzy index; supply chain; sustainability criteria

## 1. Introduction

A supply chain (SC) encompasses a network of corporations involved in various processes and activities aimed at delivering services or products to customers [1,2]. SCs add value through upstream and downstream networks, ensuring that products and services are delivered efficiently to customers. It encompasses numerous units related to supply and distribution (upstream and downstream) and the ultimate end-user [3,4]. In accordance with the International Sustainability Report, scientific researchers have recently directed their focus towards devising sustainable supply chain (SSC) schemes. The interconnections of SSCs have the potential to impact the effectiveness of global supply chain networks [5]. As customers increasingly demand sophisticated products, balancing social, economic, and environmental factors has become essential for sustainable businesses [6]. However, as multinational corporations and SCs have become more unstable and uneconomical, sustainability within the SC is once more being challenged [1,7]. Unforeseen circumstances often

interrupt industry and their SCs, further challenging SCs' sustainability. It is difficult to attain sustainability without considering persistent SC interruptions [8,9]. Hence, to attain dependable SSCs, companies must enhance and develop their capabilities for adaptability and resilience. Therefore, it is crucial to determine whether supply chains require flexibility to ensure sustainability.

Although academies have widely studied sustainability and SC concepts, sustainable and resilient SSC management (RSSCM) is yet to be fully analysed [1]. In SCs, flexibility is measured as the ability to endure and predict disruptions, react to them, and recuperate efficiently from disruptions with minimum penalties, if any [10,11]. Studies have indicated that successful SC implementation practices can mitigate or address social, economic, and environmental issues [12]. Conversely, RSSCM entails managing supplies/resources to meet stakeholders' expectations and foster high flexibility and sustainability in industries [13]. The available literature on RSSCM indicates that a procedural analysis was performed incorporating sustainability and resilient SCs, particularly in developing countries [14]. This aligns with the scarcity of inherent research in these countries. Contrary to this, amongst research outputs on the topic, Pettit et al. [15] argued that there is a necessity for supply chain sustainability that enhances the complexity of the system. Chowdhury et al. [16] highlighted the need to develop system thinking (ST) practices to tackle the increasing complexity of the system. System thinking is the ability to perceive the ecosystem as a dynamic system, where everything is interconnected, and individual actions cannot be undertaken in isolation [17–20].

Achieving RSSCM may not be feasible in isolation and might overlook the necessity for a comprehensive system evaluation. This gap in the existing literature is what the current study aims to address. The primary objective of this study is to investigate and delineate supply chain flexibility using a ranking and stationary phase approach. The anticipated outcome of this research is to provide strategic advantages and a competitive edge in the rapidly evolving construction environment, necessitating transformation. Additionally, it is expected to mitigate industry risks by enabling real-time insights into SC operations and linkages. Likewise, it is expected that construction firms will be motivated to regulate and enhance their logistics processes. Thus, the current study's overarching research question is related to the fundamental resilience enablers for SCs' sustainability. Furthermore, this signifies a shift towards RSSCM practices. Moreover, the findings derived from this study are poised to enhance the resilience and sustainability of supply chains, yielding benefits such as cost reduction and heightened manufacturing efficiency and flexibility. Consequently, this trajectory is anticipated to enhance profitability for construction firms. The concurrent implementation of resilient and sustainable SCs enables the effective management of unstable events, allowing for proactive responses and the resumption of systematic operations post-disruption.

In addition, logistics in the supply chain coordinates the shipping and storage of services and goods [21,22]. Thus, the capacity of the supply chain is essential for sustainable supply in construction projects since the process starts with raw materials and proceeds to processing, manufacturing, and distribution to the final destination [23,24]. Thus, in logistics, the capacity of the supply chain is the volume of the physical space, assets, or personnel available to carry, store or deliver products—the truck, shipping, and warehouse capacity [25,26]. Hence, a reliable and sustainable supply chain must rely on various organisations such as vendors, logistics providers, suppliers, distributors, and retailers. These must seamlessly and optimally function for sustainable constriction [27,28]. It is noteworthy that various supply chain components, including delivery on time, the availability of stock, the accuracy of orders, and production lead times, play a critical role in ensuring that supply chain performance is responsive and seamless [29,30].

Consequently, the most critical factors that influence the efficiency of the SC include ICT, data and knowledge, quality assurance, the structure of the SC, culture, record control policy, data sharing, client needs, prediction method, the length of the review period, lead time, internal incorporation support by top management, and information technology

(IT) [31]. Although these factors have been explored in the existing literature, how they affect the construction industry is still debated, even though they play a critical role in enhancing the effectiveness and performance of SCs. In the construction sector, efficient SC approaches have been applied, e.g., applying effective SC features and investing in factors that influence the SC positively and its performance [32–34]. Additionally, the ownership structure of the SC components, construction material, and various material sources can affect the SC's efficiency in the construction industry [35,36]. Therefore, a comprehensive hybrid assessment for strengthening resilience and sustainability in SCs is required for sustainable construction projects. Hence, this study theorised that hybrid assessment can strengthen resilience and sustainability in the supply chain. Based on a brief literature review, this paper describes the procedure for performing this study. Subsequently, the anticipated outcomes of the study are analysed within the existing literature's context. Finally, this research culminates by highlighting the major findings and offering suggestions for future research endeavours.

## 2. Problem Origin

Developing nations' construction firms have recorded significant changes to satisfy local economic goals [37]. Nevertheless, these countries' building companies often struggle with competitiveness due to their inadequate ability to withstand international sustainability standards. Construction projects within building cultures commonly face numerous challenges, including scheduling delays, incomplete work, cost overruns, low quality, and a significant risk of failing to achieve the intended project objectives [38,39]. Due to the limited level of investment in this sector, many enterprises ultimately find themselves either on hold or stopped [40]. Generally, the building industry in developing nations is not aligned with the goals of the authorities, society, and clients; therefore, these companies lag substantially behind compared to their counterparts [41]. Egypt is an exceedingly populated nation, with a population of ninety-five million and an annual growth of 2% [42]. Notwithstanding the variations in social conditions, the Egyptian building industry is severely hampered by the challenges listed above. Certainly, due to low wages, high unemployment, and security concerns, the construction industry is an unstable market [43]. Sudden currency variation, a lack of well-informed company decisions, and constraints in investment models contribute to the growing risk in this sector [44]. Moreover, Abd El-Razek et al. [45] discovered the major reasons for construction project delays, including financing difficulties amid project execution, owner (client) inconsistencies, unstable payments, and design amendments, and the lack of competent construction management.

Based on the preceding discussions, the significance of RSSCM is evident. RSSCM stands as a robust approach to addressing the aforementioned issues, acknowledged as a standard practice in many advanced nations. Its emphasis lies in elevating both monetary value and efficiency to jointly enhance value without conceding quality [46]. The adoption level and implementation of RSSCM are seemingly modest in emerging nations. Despite the growing demand for RSSCM implementation in these countries, the response on the ground is still insufficient to change the way the construction sector operates.

## 3. Study Ingenuity

This paper's main goal is to assist policymakers in reducing unnecessary costs and improving the quality of construction projects to achieve sustainable objectives. This is particularly beneficial for emerging nations, where there is limited understanding of the impact of operations on RSSCM stages. It is important to highlight the research gap in this field, with the building industry being no exception. It has been argued that many construction experts and investors in Egypt lack sufficient knowledge of RSSCM, hindering the implementation of RSSCM.

Consequently, the adoption of standard RSSCM practices in Egypt remains elusive. As a result, temporary measures, such as creating uncoordinated teams, are often taken that fail to reduce project costs.

Consequently, this research's primary aim is to thoroughly investigate the limited adoption of RSSCM within developing nations. In pursuit of this aim, this study is designed to accomplish the following objectives: (1) to examine the current understanding and implementation of RSSCM within the framework of sustainable construction; (2) to identify the key enablers that enhance RSSCM within the sustainable construction sector in a developing country, employing an extensive literature review; (3) to systematically categorise these influential factors through the application of brainstorming techniques; and (4) to assess and prioritize these categorized enablers using a novel hybrid measurement approach that combines alternative assessment and ranking, employing the compromise solution-fuzzy synthetic evaluation (MARCOS-FSE) method.

## 4. Literature Review

### 4.1. Sustainable Supply Chain Management (SSCM)

The current sustainable supply chain management (SSCM) systems are complex, where numerous firms should work together in different stages to supply different goods to customers [47]. To lessen the risk of interruptions and volatility and enhance the SSCM's resilience and flexibility, separate firms must cooperate [48]. SSCM encompasses all aspects of a company that are integrated into a cohesive system [49]. SSCM integrates the three dimensions of sustainability—economic, environmental, and social—throughout the manufacturing process. The product lifecycle includes designing products, sourcing raw materials, production, packaging, storage, shipment, supply, utilisation, return, and disposal [50,51]. SSCM can adeptly and efficiently handle the interconnected financial, environmental, and social aspects within global SCs [52]. To achieve sustainable supply chains, stakeholders must address economic, environmental, and social necessities [53,54]. The philosophy is that completion would be preserved by sustaining customer demands and the associated standards. SSCM has received significant recognition from amplified academic research outputs in recent years. Thus, SSCM increases value management (VM) [55]. SSCM is a methodological procedure for enhancing product value. In other words, it is a method for optimising and analysing the role of various components and their related costs to enhance the value of the product [56].

Regarding construction projects, SSCM can be significantly helpful. The early application of SSCM in construction can be cost-effective and can save time, ultimately leading to increased profits from investments and improved cost benefits [57]. SSCM enhances the adoption of cost-effective technologies and materials without compromising the functionality of the product [58]. SSCM facilitates the improvement of building supply chains by reducing costs through integrated practices while upholding high-quality deliveries, thereby enhancing sustainability [59]. Within SCM, the social aspect of sustainability has received greater attention compared to environmental and economic factors [60]. Sustainability in SCM seeks to incorporate economic, environmental, and social elements in a cost-effective manner. Sustainability in SCM is defined as a convergent target among firms within a supply chain, providing comprehensive environmental and social benefits to all links in SSCM [61,62]. It encompasses the endeavours of firms to mitigate the environmental and social impacts throughout the path of their products in SSCM, from raw material sourcing to storage, production, distribution, and ultimately to customer delivery [60,63–65].

### 4.2. Adoption of Sustainable Supply Chain Management (SSCM)

Mukherjee and Mandal [66] employed the interpretive structural modelling (ISM) approach for investigating key challenges in managing sustainability practices within the photocopier reengineering industry. The impacts of working conditions, return modes utilised, and challenges with advertising re-engineered products were found to be substantial. The factors sharing the greatest level of dependency were designing products, reengineering equipment and tools, questions regarding suitable planning of reassembly and disassembly, and the role of experience and skill of workers. By scrutinising the key en-

ablers that help to change an SC into a viable entity, a technique that effectively incorporates sustainable practices into an SC was proposed. The ISM is an interactive learning process in which elements are organised into an all-embracing system model, which is applied as the method. ISM aids in measuring the purpose and sequence of complex relationships among elements within a system [67]. A hierarchical model was constructed using the ISM approach. Elements with strong dependency and driving power were identified, including customer concerns about sustainable practices, monitoring mechanisms, the understanding of sustainable practices in SCs, and initiatives to quantify the benefits of sustainability in an SC. Grzybowska [68] described SCs' enablers of sustainability and observed how they interact. Sixteen enablers were identified, with high acceptable reverse logistics and top management practices' implementation (i.e., environmental performance) having the highest reliance and strongest driving power. Hussain [69] presented a model of diverse SC enablers, studied the different relationships, and proposed strategies for determining sustainable SCs' systems.

The concept of triple-bottom-line sustainability, encompassing economic, environmental, and social aspects, was outlined, and enablers were identified. The correlation between various sustainability aspects was established using the ISM technique, and the ISM results were incorporated into an analytical network process alongside latent alternatives to determine the most suitable substitute(s) for achieving sustainability within the SC. The more resilient and influential factors identified included customer feedback, regulatory constraints, and risk management. Diabat and Govindan [70] investigated the factors influencing the implementation of green supply chain management (GSCM) practices using an ISM methodology. The triple aspects demonstrating strong resilience and driving force are legislation and governmental regulations, green design, and reverse logistics integrating quality. The investigation of the concepts of GSCM implementation by Mathiyazhagan et al. [71] was divided into two stages: identifying barriers and a qualitative study. The ISM was utilised to comprehend the mutual effects among the twenty barriers identified through a literature review, academic research, and expert opinions.

A graded sustainable structure for assessing the barriers to GSCM adoption in a company was suggested by Dashore and Sohani [72,73]. An operational model was established using the ISM technique after identifying fourteen barriers. The barrier with the most pressing resilience power was the lack of a government inventiveness framework for GSCM suppliers' and practitioners' resilience to move towards GSCM. Muduli and Barve [74] investigated diverse behavioural factors influencing the implementation of GSCM practices and their interactions to fulfil the needs of the mining industry in India for green initiatives. The interrelationships between the identified behavioural aspects were derived via an ISM approach. The factors demonstrating greater impact and dependency were top management support and green modernisation. Luthra et al. [75,76] identified several critical factors relevant to GSCM in the Indian manufacturing industry. Using the ISM method, a contextual link among these elements was established. Among the ten standards investigated, global environmental standards and cutting-edge green applications exhibit the highest dependency and driving force.

Similarly, Kumar et al. [77] gathered primary data to prioritise different elements for successful customer engagement in the implementation of green initiatives within an SC. An ISM analysis was applied to establish a background link between the factors. Ten elements were identified for the study, with the level of consumer awareness and green labelling being found to have the highest dependence and driving force [78]. An ISM approach was used to understand how the barriers interrelate. Poor data, insufficient forecasting, and planning caused the highest reliance and strongest driving factors. Kannan et al. [79] established a multi-criteria collective decision-making model within a fuzzy framework to identify the selection of the optimal third-party reverse logistics provider (3RRLP).

The ISM procedure and the fuzzy concept of ranking based on similarity to the perfect solution were utilised for the analysis (TOPSIS). While engineering/technical proficiency was a prerequisite, the cost of reverse logistics emerged as the primary driving force.

Govindan et al. [80] employed the ISM approach for identifying and summarising the interactions among relevant factors for selecting the 3RRLP. It was determined that the attributes with the highest dependence and driving force were third-party and reverse logistics services. Applying the ISM technique, Sarkis et al. [81] investigated eleven barriers to adopting environmentally friendly manufacturing practices. The most robust dependent and driving factors were inadequate design-for-environment protocols and inappropriate evaluation methods.

Through the ISM analysis, Raut et al. [12] discovered the aspects that affect and hinder the adoption of GSCM attempts in the Nigerian building industry. They discovered that the barriers to the implementation of GSCM activities included a dearth of public awareness, insufficient understanding of ecological consequences, inadequate commitment from executive management, and insufficient legal enforcement and authority. Similarly, Balasubramanian and Shukla [82] suggested a hierarchical sustainability framework to evaluate twelve barriers to GSCM in the UAE building industry. The two factors found to have the most significant impact were identified as a shortage of resources and a lack of stakeholder awareness.

Additionally, utilising ISM, Sandeep et al. [83] outlined fifteen significant facilitators for integrating green principles into the SC of the Indian automotive industry. The primary influential factors included regulation, government policies, and comparative advantage. Eswarlal [84] utilised the ISM approach to assess fourteen crucial features of sustainable development in India related to the adoption of renewable energy. The major influential aspects and reliance identified were sustainable growth, leadership, and investment returns. This study also highlighted fourteen CSFs for the adoption of renewable energy, revealing that investment return, sustainable growth, and public awareness exhibited the highest dependency and driving force. The factors to be considered were identified, and the links between the essential elements of each functionality were measured using the ISM method. Kang and Won [85] proposed a full assessment for selecting a suitable site for establishing a wind farm. The factors that required consideration were defined, and the links between the essential elements of each functionality were measured using the ISM method. The importance of the standard was measured through a fuzzy analytic network approach while also assessing the projected overall performance of the wind farm projects.

Furthermore, Muduli et al. [86] defined the potential barriers to the implementation of environmental initiatives within India's mining sector. The two primary aspects with major reliance and a strong driving force were efficient waste management strategies and a lack of commitment from senior executives. Kholil [87] utilised the ISM approach in their research project to develop a formal model tailored to the conditions impacting coral turtles, reefs, and the diversity of pelagic fish in Bunaken Park, aiming at achieving sustainable tourism management objectives. This study identified 9 major criteria and 15 secondary criteria. The control aspects, deemed to have significant influence and greater dependence force, were measured based on setting the number of appointments and improving public commitment. Concerning the sub-criteria, the national park agency emerged as the most impactful driving force, while public, marine, and environmental non-governmental organisations played major roles. Muduli et al. [73] argued that the implementation of GSCM in the mining sector had an impact on human behaviours. These features were identified and ranked in their analysis, accompanied by ten barriers to implementing GSCM methods in the foundry business. Balaji et al. [88] used the ISM approach to define how the barriers are interrelated. The dearth of constitutional legislation and oversight had a major motivating influence, though a dearth of implementation of new tools was the most inspiring factor and had the most significant reliance.

The association among the thirteen major barriers that prevent the adoption of energy conservation in China was assessed by Wang et al. [89]. Based on the study's results, the dearth of high-tech familiarity with energy conservation had the most significant influence. Concerning Taipei Metropolitan Solid Waste Management's processes aimed at reducing air pollution, dos Muchangos et al. [90] identified eighteen criteria with significant resilience

and driving power associated with non-renewable energy or fuel usage, waste generation, and air pollution. Kumar et al. [91] identified and validated nine primary concerns for supplier selection based on corporate social responsibility (CSR) using the ISM procedure, highlighting child labour and safety procedures as having the strongest reliance and driving forces. Additionally, Mangla et al. [92] identified various performance-based standards for GSCM adoption within organisations, particularly focusing on approaches utilised by small, micro, and medium-sized enterprises (SMEs) in India. Furthermore, Mohanty et al. [93] argued that MSMEs in India face significant pressure from external investors to integrate GSCM practices.

Moreover, Muduli et al. [86] employed a combination of a literature review, a theoretic graph approach, and a matrix technique to quantitatively identify and assess the negative impact of barriers hindering the implementation of GSCM. Luthra et al. [94] evaluated the major success factors for realising environmental sustainability in India concerning the car industry. Their study revealed four expected performance measurements using factor analysis concerning GSCM implementation activities and six CSFs to implement GSCM to realise sustainability. Furthermore, the background linkages between CSFs and ranking the CSFs were analysed regarding performance indicators using the explanatory rank procedure modelling approach. Based on the results, GSCM approaches are aiding in improving environmental, economic, social, and operational performance. Many standards for improving performance in GSCM approval and adoption in India were identified and analysed by Mangla et al. [95]. The adoption of SSCM faces several obstacles, though not all exert an equal impact on sustainability initiatives.

Consequently, it is critical to find the major factors required for SSCM adoption techniques and their influence [96]. Thus, it can be inferred from the abovementioned literature that the existing literature on implementing sustainable activities relates to various countries and businesses. Research on the significance of adopting SSCM issues and techniques within the Egyptian construction sector is limited, and fewer have explored sustainable adoption activities. Hence, this demonstrates the need for additional research concerning the application of sustainable activities in Egyptian industries. The objective of this study is to categorise the various passive facilitators of resilient and sustainable supply chains in order to address this issue.

### 4.3. Major SSCM Resilience Enablers

The capacity of a supply chain to adapt against unforeseen events, referred to as resilience or flexibility, involves its ability to design for, respond to, and recover from disruptions while maintaining operational stability at an optimal level of connectivity [97]. This entails controlling both the structure and function of the SCs. Similarly, resilience can be defined as the capability to bounce back from severity [98]. Thus, a resilient supply chain is capable of enduring or mitigating the effects of interruptions and promptly recovering from them. Therefore, resilience is considered fundamental in contemporary discussions around the SCM [99,100]. Hence, a resilient supply chain can either prevent or withstand the impacts of interruptions and recover in a timely and cost-effective manner [101]. Resilience has repeatedly been proven to be a critical element in securing the success of industries. Supply chain resilience is now recognised as more than just a method for risk management [98,102]. It is therefore approved that managing risk involves being better positioned than competitors to tackle interruptions in the SCs.

Moreover, resilient supply chains provide businesses with viable advantages [103]. Thus, it is critical to understand the various aspects of resilience to develop a flexible chain. The degree of resilience is often influenced by factors such as capability, vulnerability, SC design, and orientation, which are shaped by the organisational context [104–106]. Supply chain disruptions are unforeseen events that disrupt the normal flow and operations among supply chain entities, involving products, components, and materials [107]. These disruptions can originate from various sources, including natural adversities, specific events, communication interruptions, natural hazards, terrorism, and political uncertainty [108,109].

Businesses are more prone to unexpected susceptibilities, producing insignificant interruptions across their SCs [110]. Consequently, these businesses need to identify and concentrate on their essential SC constituents, and policymakers must revaluate approaches for developing more flexible global SCs [111]. Digital tools have caused some disruptions in the building industry. As a result, project supervisors have prioritised the development of a more flexible supply chain to mitigate the impacts of disruptions [110]. SCs can address the severe effects of interruptions and significantly lessen the time required for construction industries to return to normal operations [112]. RSSCM comprises resource management to meet investors' desires to realise high sustainability and flexibility in the SCs [113,114]. The key component of RSSCM is the management of risk. Kamalahmadi and Parast [111] contended that resilience is a central component for managing SCs, which helps to promote their fast recovery from disruptions. Various techniques are employed to realise RSSCM, including the assessment of total value management, St techniques, the network perspective, and transaction costs [115,116].

In strategic network design, there are connections between supply chain resilience and sustainability performance [64,117]. Fahimnia and Jabbarzadeh [57] indicated the ways in which changes in resilience influence the environmental, economic, and social sustainability of SCs. Likewise, Ivanov [118] proposed model-based forecasting, which revealed the various ways in which SC flexibility can be connected to sustainability attributes. Correspondingly, a scenario concerning goals contradicting sustainability and resilience was evaluated [119].

Conversely, safeguarding a facility requires the simultaneous enhancement of sustainability and resilience. In contrast, the safeguarding of facilities must be performed concurrently to enhance sustainability and resilience [120]. Building upon the fundamental principles of SCM and RSSCM within construction projects, this research highlights the fundamental resilience enablers in SSCM (Table 1).

**Table 1.** Major SSCM resilience enablers.

| Enablers | Code | References |
|---|---|---|
| Adaptability | EBL1 | [15,121] |
| Agility | EBL2 | [122,123] |
| Collaboration | EBL3 | [15,107] |
| Compatibility | EBL4 | [1,122] |
| Composure | EBL5 | [124] |
| Contingency Planning | EBL6 | [125] |
| Corporate Social | EBL7 | [126] |
| Flexible Structure | EBL8 | [124,127] |
| Flexible Transportation | EBL9 | [125] |
| Health | EBL10 | [122,123] |
| Information Security | EBL11 | [128] |
| Information Sharing | EBL12 | [129,130] |
| Just in Time | EBL13 | [106,116] |
| Leadership | EBL14 | [1,15,107,121] |
| Market Sensitivity | EBL15 | [129,130] |
| Proper Scheduling | EBL16 | [15,107] |
| Quality Awareness | EBL17 | [125] |
| Resource Efficiency | EBL18 | [122,123] |
| Responsibility | EBL19 | [127,131] |
| Responsiveness | EBL20 | [124,125] |
| Risk and Revenue Sharing | EBL21 | [132,133] |
| Risk Management Culture | EBL22 | [124,125] |
| Safety Stock | EBL23 | [124] |
| Self-Regulation | EBL24 | [125] |
| Strategic Risk Planning | EBL25 | [106,116] |
| Supply Chain Security | EBL26 | [123] |
| Swift Trust | EBL27 | [121,122] |

**Table 1.** *Cont.*

| Enablers | Code | References |
|---|---|---|
| Technological Capability | EBL28 | [124,125,134] |
| Tenacity | EBL29 | [15,121] |
| Top Management Support | EBL30 | [1,122] |
| Transparency | EBL31 | [124,125,134] |
| Visibility | EBL32 | [1,15,107,121] |

## 5. Research Methods

This study's research methodology began with an extensive literature review, as shown in Figure 1, to identify the key enablers of RSSCM. The flowchart comprises four steps: extensive literature review, brainstorming, pilot survey, questionnaire administration, and data analysis using fuzzy semantic evaluation (FSE). Consequently, a wide-ranging search through the scholarly archives, comprising Web of Science and Scopus, was performed, compassing studies from 1995 to 2022, ultimately extracting 32 critical enablers of RSSCM. According to Ali et al. [135], these databases are the most used databases in extensive reviews. Moreover, brainstorming was employed to classify and group a list of enablers. This technique fosters creativity, collaboration, idea generation, problem-solving, and information organisation. Brainstorming also expedites categorising variables, ensuring comprehensive, well-structured outcomes [135]. A preliminary survey was subsequently carried out to ensure the effectiveness and quality of the research process. This pilot survey served as a crucial step to measure the viability, lucidity, and profundity of the survey tool designed for the main study. By conducting the pilot survey, the research methodology aimed to validate that the study's objectives were accomplished and that the survey instrument was well structured and ready for implementation in the larger research effort [136].

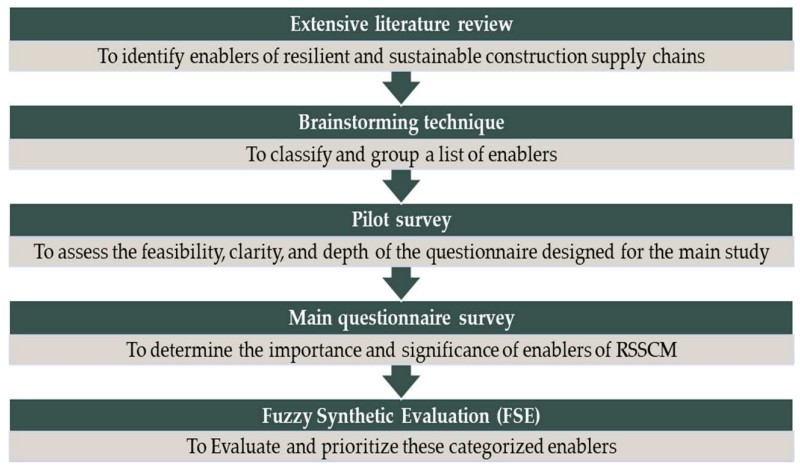

**Figure 1.** Study flowchart.

Therefore, a survey tool was created to measure the significance of SSCM enablers. Respondents from Cairo and Giza—the regions home to most of Egypt's development enterprises—were chosen to represent the storm drainage industry. The survey tool was categorised into three parts, the first of which aimed to collect data on the respondent's background information and experience with SSCs. The open-ended questions in the following sections were designed to cover any requirements that participants thought were crucial for enabling RSSCM. Using a five-point Likert scale based on their level of experience and competency, respondents ranked each criterion. Numerous investigations and fields of study have made extensive use of this scale [137–144].

This research employed a probability random sampling technique, giving each Giza and Cairo expert an equivalent chance of being nominated. It has been argued that probability random sampling is a widely recognised approach in social science studies.

Probability sampling is a technique applied in construction research to select a survey sample from a much larger sample size so that each element or individual in the known population has a non-zero probability of being selected [145,146]. Thus, this ensures that the sample is representative of the larger population, enabling justifiable conclusions or generalisation from the data set. There are a variety of probability sampling techniques, including simple random sampling, systematic, stratified, and cluster sampling [147,148]. Simple random sampling has been successfully applied in prior research studies, including Ali et al. [149], to select their intended participants, substantiating its reliability within these study types. Additionally, the study's goal also played a role in determining the sample size [150]. It would be adequate to have more than thirty cases for graphic (descriptive) analyses, e.g., the mean, median, and mode for a regular distribution curve [151]. A good questionnaire survey response rate, or rate of return, is above average, and based on some industry data, it should be above at least 25% and should be subject to adequate total responses [152,153]. In this study, a response rate of 60% was achieved. This is deemed acceptable based on the existing literature [154,155]. Consequently, one hundred survey forms were dispersed, and sixty forms were filled out and returned. Based on the rate of return principle, this accounted for 60%, deemed adequate for further analysis [156,157].

Our research employed a hybrid MARCOS-FSE technique to assess the factors influencing SSCM adoption. This study significantly contributes to decision-making by introducing an innovative approach designed to aid decision-makers in addressing intricate problems. Moreover, this method provides an extensive decision support system capable of addressing diverse challenges across various domains. The core of the MARCOS approach lies in ranking and assessing alternatives using a compromise solution methodology. This involves establishing utility functions based on the distance between ideal and anti-ideal solutions and their aggregations [158,159].

Moreover, FSE utilises fuzzy logic for assessing multi-criteria decision processes. It is an artificial intelligence technique that quantifies subjective judgments, mitigating uncertainty and vagueness, thus enhancing objectivity. FSE employs mathematical calculations to convert linguistic criteria into quantifiable data, ensuring precise analysis. Using fuzzy logic, FSE overcomes binary logic's limitations in handling imprecision and uncertainty [160]. This study employs a quantitative questionnaire survey, which inherently entails uncertainty and vagueness in respondents' judgments. In this context, the choice of FSE assists authors in addressing this inherent vagueness, facilitating a systematic evaluation of enabler categories and aiding in identifying the optimal option based on predefined criteria.

## 6. Results

The results indicate that brainstorming enabled the grouping of the enablers into four distinct categories, providing a structured framework for understanding and organising them. This makes it easier to analyse and work with the enablers in the context of this study. The integration of MARCOS and FSE offered a robust decision-making approach, proposing a resilient and comprehensive decision-support system capable of tackling intricate real-world issues. Thus, this study employed this approach and assessed and prioritised the key determinants enabling RSSCM.

### 6.1. Brainstorming

Brainstorming has led to the categorisation of the enablers into four distinct categories. This categorisation provides a structured framework for understanding and organising the enablers, making it easier to analyse and work with them in the context of this research study. Categorising enablers helps to identify common themes or relationships among them, which can be valuable for drawing insights and conclusions from the data. Figure 2 demonstrates each category along with their corresponding enablers.

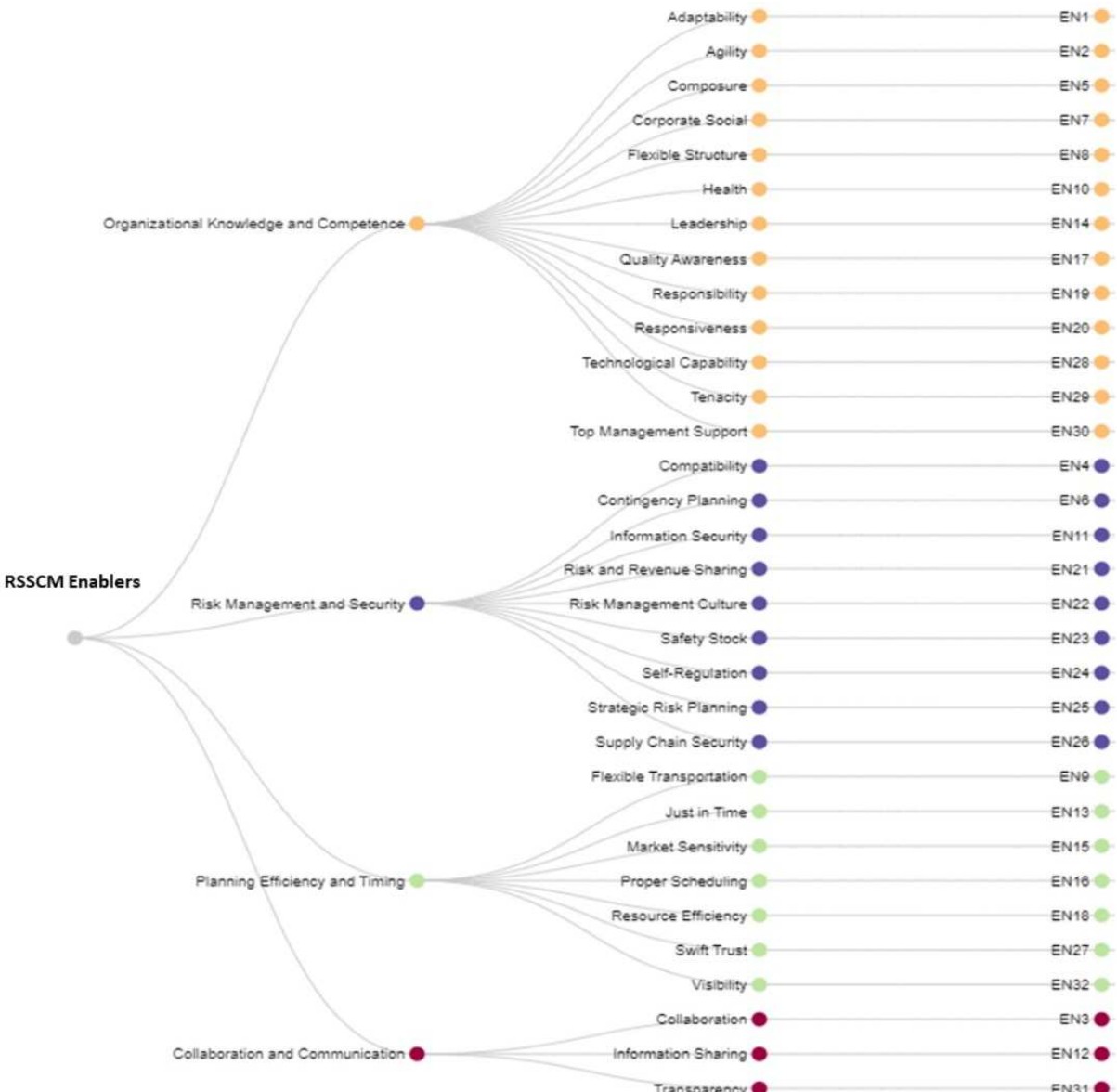

**Figure 2.** Categorisation of the RSSCM enablers.

### 6.2. Hybrid MARCOS-FSE Method

The integration of MARCOS and FSE constitutes a robust decision-making method, offering a resilient and all-encompassing decision-support system capable of tackling intricate real-world issues. This study employs this approach to assess and prioritise the key determinants enabling RSSCM.

In MARCOS, the initial phase involves creating a decision matrix and pinpointing the optimal alternative (OA) and the worst alternative (WA) using Equations (1) and (2). In this research, these alternatives represent the elements facilitating RSSCM adoption in the construction sector, whereas the criteria embody the evaluations provided by experts utilising a Likert scale. Additionally, according to each expert's viewpoint, the highest and lowest performance measures are those that represent the best and worst options.

$$OA = \begin{cases} min(x_{ij}), & for\ cost\ criterion \\ max\ (x_{ij}), & for\ benefit\ criterion \end{cases} \tag{1}$$

$$WA = \begin{cases} max(x_{ij}), & for\ cost\ criterion \\ min(x_{ij}), & for\ benefit\ criterion \end{cases} \tag{2}$$

where $x_{ij}$ represents the performance measure corresponding to the $i$th alternative related to the $j$th criterion.

Considering the "EBL1-Adaptability", the subsequent stage involves constructing the normalised decision matrix ($\delta_{ij}$) using Equation (3). This process is iterated for the remaining factors that enable supply chain adoption.

$$\varrho_{ij} = \begin{cases} \dfrac{x_{aij}}{x_{ij}}, & for\ cost\ criterion \\ \dfrac{x_{ij}}{x_{aij}}, & for\ benefit\ criterion \end{cases} \tag{3}$$

$$\varrho_{ij} = \left[ \left(\frac{5.0}{5.0}\right) + \left(\frac{5.0}{5.0}\right) + \left(\frac{4.0}{5.0}\right) + \left(\frac{4.0}{5.0}\right) + \left(\frac{5.0}{5.0}\right) + \left(\frac{4.0}{5.0}\right) \right] + \cdots = 49.53$$

An equal importance weight of 1.69% is assigned to each of the 59 responses. The third step includes developing a weighted decision matrix ($\beta_{ij}$) that can be computed based on Equation (4).

$$\beta_{ij} = \varrho_{ij} \times w_j \tag{4}$$

$$\beta_{ij} = 1.69\% \times 49.53 = 0.837$$

After that, the utility functions ($f(\vartheta_i)$) can be computed as Equation (5).

$$f(\vartheta_i) = \frac{\vartheta_i^+ + \vartheta_i^-}{1 + \frac{1 - f(\vartheta_i^+)}{f(\vartheta_i^+)} + \frac{1 - f(\vartheta_i^-)}{f(\vartheta_i^-)}} \tag{5}$$

$$f(\vartheta_i) = \frac{0.837 + 1.604}{1 + \frac{1 - 0.657}{0.657} + \frac{1 - 0.343}{0.343}} = 0.710$$

where $\vartheta_i^+ = \frac{S_i}{S_{BA}}$ and $\vartheta_i^- = \frac{S_i}{S_{WA}}$ represent the utility degrees concerning the optimum and worst alternatives, respectively. It is worth noting that $S_i = \sum_{i=1}^n \beta_{ij}$. Additionally, $f(\vartheta_i^+) = \frac{\vartheta_i^-}{\vartheta_i^+ + \vartheta_i^-}$ and $f(\vartheta_i^-) = \frac{\vartheta_i^+}{\vartheta_i^+ + \vartheta_i^-}$ represent the utility functions of the optimum and worst alternatives, respectively. The utility functions of alternatives, ranging between 0 and 1, signify the relative significance of different enablers of RSSCM within the construction sector. Table 2 demonstrates the utility function scores (UFS) corresponding to the RSSCM enablers in this industry.

In Table 3, the evaluation of RSSCM enablers using FSE involves the computation of the total mean values (*Total MS*$_{Category}$) and overall mean values (*Overall MS*), as well as the weight for each enabler ($W_{enabler}$) and category ($W_{Category}$) based on the enablers' mean score $MS_i$. These computations are based on the following equations.

$$Total\ MS_{Category} = \sum_i^{Enablers\ in\ this\ categroy} MS_i \tag{6}$$

$$Overall\ MS = \sum_{Category}^{All\ Categories} Total\ MS_{Category} \tag{7}$$

$$W_{enabler} = \frac{MS_i}{Total\ MS_{Category}} \tag{8}$$

$$W_{Category} = \frac{Total\ MS_{Category}}{Overall\ MS} \tag{9}$$

The levels of membership functions (*MF*) for every enabler and category are established. The level of an element's membership in a fuzzy collection is indicated by its *MF*. Calculating $MF_{En_{in}}$ for the major enablers (level 3) using Equation (10) is a prerequisite for analysing the enabler categories. The *MF* level 3 values are presented in Table 4, forming the basis for generating the *MF* level 2 for each category. Equation (11) is used to create the level 2 *MFs* for category (*Di*), and it requires the weights linked to enablers. The results of *MF* level 2 are displayed in Table 5. In Equation (12), the fuzzy matrix (*Ri*) of enablers is used to compute the membership functions (*MFs*), while Equation (13) represents the calculation of *Di*.

Furthermore, the *MF* at level 1 for categorised enablers can be determined using the same method, with Table 6 revealing the results of *MF* level 1. Lastly, Equation (14) is used to compute the overall level (*OL*) for each category. This can be achieved by incorporating the increments' findings and the three *MF* levels. The values of *OL* are presented in Table 7, which, in turn, serves as the basis for determining the ranks for each category.

$$MF_{En_{in}} = \frac{En_{1_{in}}}{L_1} + \frac{En_{2_{in}}}{L_2} + \frac{En_{3_{in}}}{L_3} + \frac{En_{4_{in}}}{L_4} + \frac{En_{5_{in}}}{L_5} \tag{10}$$

$$D_i = W_i \otimes R_i \tag{11}$$

$$R_i = \begin{vmatrix} MF_{En_{i1}} \\ MF_{En_{i2}} \\ \dots \\ MF_{En_{in}} \end{vmatrix} = \begin{vmatrix} En_{1_{i1}}\ En_{2_{i1}}\ En_{3_{i1}}\ En_{4_{i1}}\ En_{5_{i1}} \\ En_{1_{i2}}\ En_{2_{i2}}\ En_{3_{i2}}\ En_{4_{i2}}\ En_{5_{i2}} \\ \dots\ \dots\ \dots\ \dots\ \dots \\ En_{1_{in}}\ En_{2_{in}}\ En_{3_{in}}\ En_{4_{in}}\ En_{5_{in}} \end{vmatrix} \tag{12}$$

$$D_i = W_i \otimes R_i = (W_1, W_2\dots, W_n) \otimes \begin{vmatrix} En_{1_{i1}}\ En_{2_{i1}}\ En_{3_{i1}}\ En_{4_{i1}}\ En_{5_{i1}} \\ En_{1_{i2}}\ En_{2_{i2}}\ En_{3_{i2}}\ En_{4_{i2}}\ En_{5_{i2}} \\ \dots\ \dots\ \dots\ \dots\ \dots \\ En_{1_{in}}\ En_{2_{in}}\ En_{3_{in}}\ En_{4_{in}}\ En_{5_{in}} \end{vmatrix} = (d_{i1}, d_{i2}, \dots, d_{in}) \tag{13}$$

$$OL = \sum_{i=1}^{n} D_{C_i} \times L_i \tag{14}$$

where $En_{1_{in}}$, $En_{2_{in}}$, $En_{3_{in}}$, $En_{4_{in}}$, and $En_{5_{in}}$ are the percentages of respondents who evaluated the significance of a specified barrier from 1 to 5. By using a five-point Likert scale, $L_1$, $L_2$, $L_3$, $L_4$, and $L_5$ were provided. Additionally, $d_{in}$ refers to the degree of membership. Finally, $D_{C_i}$ is a second-level *MF* fuzzy matrix for each category and $L_i$ is the Likert scale.

**Table 2.** Prioritised factors enabling RSSCM in construction projects.

| Factors | UFS | Rank | Factors | UFS | Rank |
|---|---|---|---|---|---|
| EBL1 | 0.710138 | 2 | EBL17 | 0.6857 | 14 |
| EBL2 | 0.649762 | 24 | EBL18 | 0.69145 | 9 |
| EBL3 | 0.687138 | 12 | EBL19 | 0.600886 | 32 |
| EBL4 | 0.628918 | 28 | EBL20 | 0.67995 | 15 |
| EBL5 | 0.689294 | 11 | EBL21 | 0.644012 | 26 |
| EBL6 | 0.702951 | 5 | EBL22 | 0.602324 | 31 |
| EBL7 | 0.710138 | 2 | EBL23 | 0.694325 | 7 |
| EBL8 | 0.687138 | 12 | EBL24 | 0.639699 | 27 |
| EBL9 | 0.69145 | 9 | EBL25 | 0.603761 | 30 |

**Table 2.** *Cont.*

| Factors | UFS | Rank | Factors | UFS | Rank |
|---|---|---|---|---|---|
| EBL10 | 0.664137 | 22 | EBL26 | 0.671325 | 18 |
| EBL11 | 0.702951 | 6 | EBL27 | 0.6627 | 23 |
| EBL12 | 0.609511 | 29 | EBL28 | 0.704388 | 4 |
| EBL13 | 0.6742 | 16 | EBL29 | 0.646168 | 25 |
| EBL14 | 0.692888 | 8 | EBL30 | 0.6742 | 16 |
| EBL15 | 0.669887 | 20 | EBL31 | 0.667012 | 21 |
| EBL16 | 0.723076 | 1 | EBL32 | 0.671325 | 18 |

**Table 3.** Results for FSE of mean values and weightings for enablers and categories.

| Categories | Enablers Code | Rank | $MS_i$ | Total MS | Overall MS | $W_i$ | $W_{Category}$ |
|---|---|---|---|---|---|---|---|
| Organisational Knowledge and Competence | EBL1 | 26 | 2.932 | 39.983 | | 0.073 | 0.408 |
| | EBL2 | 9 | 3.186 | | | 0.080 | |
| | EBL5 | 18 | 3.068 | | | 0.077 | |
| | EBL7 | 32 | 2.644 | | | 0.066 | |
| | EBL8 | 16 | 3.085 | | | 0.077 | |
| | EBL10 | 28 | 2.814 | | | 0.070 | |
| | EBL14 | 24 | 2.949 | | | 0.074 | |
| | EBL17 | 8 | 3.203 | | | 0.080 | |
| | EBL19 | 19 | 3.068 | | | 0.077 | |
| | EBL20 | 3 | 3.288 | | | 0.082 | |
| | EBL28 | 11 | 3.136 | | | 0.078 | |
| | EBL29 | 4 | 3.288 | | | 0.082 | |
| | EBL30 | 1 | 3.322 | | | 0.083 | |
| Risk Management and Security | EBL4 | 13 | 3.136 | 27.576 | 97.983 | 0.114 | 0.281 |
| | EBL6 | 25 | 2.949 | | | 0.107 | |
| | EBL11 | 15 | 3.102 | | | 0.112 | |
| | EBL21 | 20 | 3.051 | | | 0.111 | |
| | EBL22 | 2 | 3.322 | | | 0.120 | |
| | EBL23 | 30 | 2.712 | | | 0.098 | |
| | EBL24 | 17 | 3.085 | | | 0.112 | |
| | EBL25 | 6 | 3.220 | | | 0.117 | |
| | EBL26 | 23 | 3.000 | | | 0.109 | |
| Collaboration and Communication | EBL3 | 5 | 3.254 | 9.153 | | 0.356 | 0.093 |
| | EBL12 | 10 | 3.186 | | | 0.348 | |
| | EBL31 | 31 | 2.712 | | | 0.296 | |
| Planning Efficiency and Timing | EBL9 | 7 | 3.220 | 21.271 | | 0.151 | 0.217 |
| | EBL13 | 14 | 3.119 | | | 0.147 | |
| | EBL15 | 21 | 3.051 | | | 0.143 | |
| | EBL16 | 22 | 3.017 | | | 0.142 | |
| | EBL18 | 27 | 2.932 | | | 0.138 | |
| | EBL27 | 29 | 2.780 | | | 0.131 | |
| | EBL32 | 11 | 3.153 | | | 0.148 | |

**Table 4.** Results for FSE of *MF* Level 3.

| Enablers Code | MF Level 3 | | | | |
|---|---|---|---|---|---|
| EBL1 | 0.051 | 0.525 | 0.000 | 0.288 | 0.136 |
| EBL2 | 0.068 | 0.407 | 0.000 | 0.322 | 0.203 |
| EBL5 | 0.068 | 0.441 | 0.000 | 0.339 | 0.153 |
| EBL7 | 0.102 | 0.576 | 0.000 | 0.220 | 0.102 |
| EBL8 | 0.085 | 0.424 | 0.000 | 0.305 | 0.186 |
| EBL10 | 0.085 | 0.525 | 0.000 | 0.271 | 0.119 |
| EBL14 | 0.085 | 0.458 | 0.000 | 0.339 | 0.119 |
| EBL17 | 0.068 | 0.390 | 0.000 | 0.356 | 0.186 |
| EBL19 | 0.034 | 0.475 | 0.000 | 0.373 | 0.119 |
| EBL20 | 0.068 | 0.356 | 0.000 | 0.373 | 0.203 |
| EBL28 | 0.051 | 0.458 | 0.000 | 0.288 | 0.203 |
| EBL29 | 0.085 | 0.373 | 0.000 | 0.254 | 0.288 |
| EBL30 | 0.085 | 0.339 | 0.000 | 0.322 | 0.254 |
| EBL4 | 0.068 | 0.407 | 0.000 | 0.373 | 0.153 |
| EBL6 | 0.102 | 0.458 | 0.000 | 0.271 | 0.169 |
| EBL11 | 0.068 | 0.424 | 0.000 | 0.356 | 0.153 |
| EBL21 | 0.068 | 0.424 | 0.000 | 0.407 | 0.102 |
| EBL22 | 0.051 | 0.373 | 0.000 | 0.356 | 0.220 |
| EBL23 | 0.102 | 0.525 | 0.000 | 0.305 | 0.068 |
| EBL24 | 0.085 | 0.407 | 0.000 | 0.356 | 0.153 |
| EBL25 | 0.102 | 0.356 | 0.000 | 0.305 | 0.237 |
| EBL26 | 0.051 | 0.492 | 0.000 | 0.322 | 0.136 |
| EBL3 | 0.068 | 0.356 | 0.000 | 0.407 | 0.169 |
| EBL12 | 0.085 | 0.407 | 0.000 | 0.254 | 0.254 |
| EBL31 | 0.085 | 0.542 | 0.000 | 0.322 | 0.051 |
| EBL9 | 0.085 | 0.339 | 0.000 | 0.424 | 0.153 |
| EBL13 | 0.085 | 0.390 | 0.000 | 0.373 | 0.153 |
| EBL15 | 0.102 | 0.407 | 0.000 | 0.322 | 0.169 |
| EBL16 | 0.051 | 0.475 | 0.000 | 0.356 | 0.119 |
| EBL18 | 0.102 | 0.458 | 0.000 | 0.288 | 0.153 |
| EBL27 | 0.102 | 0.508 | 0.000 | 0.288 | 0.102 |
| EBL32 | 0.068 | 0.424 | 0.000 | 0.305 | 0.203 |

**Table 5.** Results for FSE of *MF* Level 2.

| Category | MF Level 2 | | | | |
|---|---|---|---|---|---|
| Organisational knowledge and Competence | 0.042 | 0.258 | 0.000 | 0.185 | 0.105 |
| Risk Management and Security | 0.045 | 0.252 | 0.000 | 0.200 | 0.092 |
| Collaboration and Communication | 0.046 | 0.253 | 0.000 | 0.194 | 0.097 |
| Planning Efficiency and Timing | 0.053 | 0.222 | 0.000 | 0.222 | 0.093 |

**Table 6.** Results for FSE of *MF* Level 1.

| Category | MF Level 1 | | | | |
|---|---|---|---|---|---|
| Overall levels for all categories | 0.046 | 0.248 | 0.000 | 0.198 | 0.098 |

**Table 7.** Results for FSE of *OL* and final rank.

| Component | OL | Rank |
|---|---|---|
| Organisational Knowledge and Competence | 1.822 | 2 |
| Risk Management and Security | 1.813 | 3 |
| Collaboration and Communication | 1.811 | 4 |
| Planning Efficiency and Timing | 1.849 | 1 |

## 7. Discussion

Relying on the results of the MARCOS-FSE, the ranks of the enabler categories, in descending order, are as follows: Planning Efficiency and Timing, Organisational Knowledge and Competence, Risk Management and Security, Collaboration, and Communication. Each category will be discussed separately, highlighting its significance.

### 7.1. Planning Efficiency and Timing

This category appears to be the most significant enabler category, ranking at the top with an *OL* of 1.849. This suggests that focusing on Planning Efficiency and Timing is crucial for the success of SCSCs. There tends to be a significant correlation between timing, effective planning, and SSCs in the building industry. An SSC in construction aims to lessen the construction sector's objectionable environmental and social consequences while offering high-quality construction services and reducing material costs. Timing and efficient planning are critical to realising these objectives. One major component of timing and efficiency for SCC in construction is evaluating and identifying social and environmental opportunities and risks [161]. There is a need to accept the possible influences of construction activities on the environment and society and the capacity to evaluate and identify alternative technologies, materials, and processes that can lessen these effects [162]. Effective planning and timing could assist in recognising prospects to enhance resource usage, waste reduction, and sustainability all over the SC. Another component in preparing sustainable construction SCs is supporting objectives and goals between participants. There is a need for communication and collaboration between participants, including designers, contractors, manufacturers, and suppliers [163]. Efficient planning and timing could aid in recognising common objectives and goals between participants and establishing strategies for realising them. Generally, timing and efficiency are central to establishing and adopting SSCs in construction. Efficient timing and planning can assist in recognising opportunities and risks, lining up objectives and goals, and establishing approaches for planning and enhancing sustainability through SCs. Thus, this can lead to more efficient utilisation of resources, waste reduction, and lessening the environmental impact of construction projects [164].

### 7.2. Organisational Knowledge and Competence

The "Organisational Knowledge and Competence" category ranks second among the categorised enablers, with an *OL* of 1.822. This category is pivotal for an organisation's success. Organisational knowledge includes collective wisdom, information, and explicit and tacit expertise, from best practices to employee insights. Competence reflects the ability to apply this knowledge, involving skills, capabilities, and translating knowledge into actions. The interplay between these factors fosters adaptability, informed decisions, innovation, and organisational excellence. A significant correlation can be observed between sustainable construction supply chains (SCSCs) and the company's competence

and knowledge. SCSCs aim to reduce the building industry's harmful environmental and social impacts while enhancing high-quality, economical construction services and materials. A major component of SCSCs is innovative, ecologically friendly technologies and materials [165]. The latest knowledge concerning advances in construction sustainability and the capacity to apply them in the SC is needed. Moreover, adopting sustainable activities requires the participation of many stakeholders, including designers, contractors, manufacturers, and suppliers [82]. Technical compatibility with major buyers and suppliers is related to environmental monitoring and collaboration. Regarding logistics integration, only the environmental compatibility of suppliers with improved major suppliers showed a correlation [166]. Using the diffusion of innovation theory regarding the implementation of a specific product's SC, it was found that the exporters' implementation of a specific product's SC is not exclusively regulated by apparent comparative advantage and compatibility, but also by factors like awareness and complexity [167]. Generally, knowledge is a must for implementing and developing SCSCs [168]. Effective knowledge management between participants can enhance best practices and innovation, resulting in more viable construction practices and lessening environmental effects. Sustainable supply chain management can be improved through a generic planning approach by integrating methods and concepts to address sustainability issues [51]. Thus, future sustainable supply chain management planning requires closing the gaps between theory and practice. A critical assessment of the present supply chain planning theory offers very limited findings that are of practical importance. Supply chain planning is not offered as an intervention, and the findings are not presented in a manner that is operational for practitioners. The literature is nearly non-existent regarding tackling concerns based on context; it offers limited proof of intended results and has failed to identify the unplanned results [169]. Therefore, research is unavailable to bolster the theoretical underpinnings of how anticipated and unanticipated results are achieved. Future research agendas must leverage the understanding of the enabling mechanisms to recommend research to make mature supply chain planning implementable [169].

## 7.3. Risk Management and Security

"Risk Management and Security" ranks third among the categorised enablers, with an *OL* equal to 1.813. In SCSCs, security measures and risk management are pivotal in maintaining social and environmental sustainability while ensuring the supply chain's reliability. When integrated effectively, these elements are highly compatible with sustainability objectives [170]. Contingency planning helps to prepare for disruptions and sustain project schedules, while information security safeguards critical data used in environmentally responsible decision-making. Risk and revenue sharing incentivises sustainability among stakeholders, fostering collective responsibility [161]. A risk management culture ensures awareness of environmental and social risks, while self-regulation enables companies to exceed legal standards voluntarily. Safety stock can be employed to buffer eco-friendly materials, reducing the need for last-minute substitutions [171]. Strategic risk planning aligns risk management with long-term sustainability goals, considering factors like climate change and evolving regulations. Supply chain security, guarding physical assets and supply chain integrity, is crucial in preventing disruptions with far-reaching environmental and social implications. Ultimately, the compatibility of these elements promotes sustainability and resilience within construction supply chains. The role of risk management in supply chain management is to detect, examine, and offer solutions for accountability, as well as monitor and control risks [172]. The current results are comparable to the existing literature. For example, an assessment of sustainable chain risk management using the integrated fuzzy TOPSIS-CRITIC method revealed seven criteria and 44 sub-criteria, and the most dominant sub-criteria were related to equipment and machine risks, major supply failures, fluctuating demand, IT security, government policy risks, economic concerns, and a dearth of sustainable environmental management [172]. Hence, a country's sustainability risk can be used to inform sustainable supply chain management [173]. Skilful planning and risk

control of the continuity of the company's operation positively affect the company's value, image, and ability to realise the planned objectives in the social, environmental, and economic dimensions [174]. The current results concur with a new plithogenic TOPSIS-Critic model for sustainable supply chain risk management (SSRCM) proposed by Abdel-Basset and Mohamed [175]. Additionally, the results indicate the significance of each criterion in assessing SSRCM.

*7.4. Collaboration and Communication*

"Collaboration and Communication" ranks fourth among the categorised enablers, with an *OL* of 1.811. The implementation of collaborative actions of innovative and sustainable SCs indicated positive correlations among supportive activities and programs for green certification in small and medium businesses and larger businesses [176]. Collaboration and knowledge sharing among these participants were critical to guaranteeing that sustainability activities are adopted throughout the SC [177]. This differs from the common idea that the size of organisations does not restrain the association among sovereign cooperative activities and viable green building actions.

Notwithstanding this, other connections were buttressed by the analytical model. Therefore, the size of companies partially moderated the connections between cooperative activities, SCSC implementation activities, and environmental performance. Hence, adoption activities are central to promoting cooperative actions with suppliers and vendors for an innovative SSC. Lastly, they contribute to inventive SSC practice viability through collaborative actions in the supply chain process [176]. The current findings could aid in decision-making for communicative and collaborative planning in sustainable supply chain management. Companies have to deal with rising concerns in their supply chains. Thus, decision-makers must establish a balanced economic performance with social and environmental concerns. Establishing concepts and methods for jointly optimising supply chains' social, economic, and environmental operational costs is challenging [178]. Since the theory of building initiatives with implications for the conceptualisation of sustainable supply chain management is lacking in the literature, this study was able to identify how collaboration and communication can influence sustainable supply chain management in construction projects by identifying trends and how collaboration and communication can be used to increase the sustainability of construction projects. Strategic collaboration and SSCM play a mediating role in external and internal data sharing [179].

## 8. Conclusions and Recommendations

In various developed and developing nations, fostering vibrant and resilient SSCs is a considerable challenge. Resilience is key for businesses' ability to realise sustainability in the contemporary unstable world markets. Establishing a more resilient and sustainable SC scheme in this analysis indicated the key enablers of RSSCM. Thirty-two enablers were obtained using the literature. Consequently, data were gathered from the respondents in the construction industry in emerging nations. To assist in addressing this issue, novel enablers are required to be adopted to select resilient and sustainable SCs. Therefore, this study focuses on this issue. Resilient and sustainable SCs were assessed by obtaining 32 enablers. Brainstorming enabled the grouping of the enablers into four distinct categories, providing a structured framework for understanding and organising them. The integration of MARCOS and FSE offered a robust decision-making approach, proposing a resilient and comprehensive decision-support system capable of tackling intricate real-world issues. Thus, this study employed this approach and assessed and prioritised the key determinants enabling RSSCM. Thus, the MARCOS-FSE technique was used to evaluate and rank the enablers. This research contributed to the available literature by adding valuable findings that might help to improve the understanding of resilient and sustainable SCs.

Similarly, it plays a critical role in future research in this field. Based on the results, it is suggested that companies increase these enablers and offer prospects for experts to improve their capabilities for sustainable adoption. Training, workshops, and seminars can

assist participants in understanding the principles underlying the topic, despite hands-on experience with the competency itself, which can help them to comprehend these specifics. This research used a quantitative survey tool to discover 32 enablers for evaluating resilient and sustainable SCs in Cairo and Giza, Egypt. These results might also help Egypt's businesses become more environmentally friendly. These findings might help organisations to manage SC interruptions via procurement inputs from a resilient supply chain base, allowing them to substitute suppliers when manufacturing is threatened. There is a dearth of studies using the ranking and stationary technique for comparable objectives. Thus, the approach taken in this study is pioneering, as it is the first to address the complexity within the construction sector in developing countries, aiming to promote RSSCM.

## 9. Theoretical and Practical Implications

This research provides various practical and theoretical implications that could be applied within the construction industry as well as in academic settings. The stagnation of construction project delivery in Egypt can be linked to the idea that they are still performed via traditional, outdated approaches. Likewise, it might be linked to a general averseness to accept modernisation. It is important to make these modifications, though stakeholders may be required to approve novel substitute designs, particularly those that affect the actual delivery of the project. The current results further reveal that construction firms in Egypt have not used SSCs to support their implementation. Stakeholders need to recognise and adopt innovative designs through seminars and workshops to ensure efficient delivery of projects. This will simplify the clients' concerns and clarify their misinterpretation of rising costs. These findings could further help clients and managers eliminate and recognise the essential barriers to implementing sustainable and resilient supply chains. Experts in the SC area must learn the ideas, methods, and principles outlined in environmentally approachable manners.

Moreover, organisations in Egypt with a role in SCs have to provide regular workshops and seminars for their personnel and integrate the role of these activities into their regular staff performance appraisal for development and growth. The government plays a crucial role in implementing public projects and developing and promoting policies and regulations across a broader industrial scale. Hence, the government must work to establish rules and regulations that could enable the application of enablers of SSCs in Egyptian industries.

Concerns about the environment are an integral component of industry planning strategy due to stringent ecological and government restraints and the prospects of ecological responsibility. Sustainability thinking is currently essential due to its significance for the industries in Egypt, which consider sustainability's ecological, social, and economic components. Recently, industries in Egypt have achieved notable steps in ecological conservation, safety, and social accountability. However, there is more room for enhancement. The components of pollution management, conserving biodiversity, and global environmental change must be prioritised consistently by industries in Egypt. Effective partnerships with other shareholders or stakeholders and the level of commitment by companies will influence how the social accountability creativities perform. The problems are becoming more and more complex, and future success will not depend only on technical solutions. Likewise, companies will need the capacity to negotiate with different partners, including public and private institutions [180]. The current findings contribute to sustainable development in Egypt. To meet the societal demands concerning industrial safety and budget until substitute energy sources are accessible by offering the needed funds, training, and technology, this type of study is required. Specifically, the results obtained from this investigation will have both theoretical and practical implications.

### 9.1. Theoretical Implications

This study has shown our novel attempt to employ hybrid assessment for strengthening resilience and sustainability in the supply chain. This study revealed that brainstorming

enabled the grouping of the enablers into four distinct categories, which provides a structured framework for understanding and organising them. Theoretically, the results obtained will enrich the literature as follows:

1. Theoretically, this research lays the foundation for employing hybrid assessment to strengthen resilience and sustainability in the supply chain of a developing country.
2. Similarly, the integration of MARCOS and FSE offered a robust decision-making tool, which could enrich the literature on proposing a resilient and comprehensive decision-support system capable of tackling intricate real-world issues.
3. This study's findings can further enrich the literature on how MARCOS and FSE can be used to identify SSCM enablers and overall supply chain management in the construction industry.

*9.2. Practical Implications*

This study's results could have some practical implications.

1. The results could be used to shrink the negative impacts of building processes on the environment.
2. SSCM could allow for effective collaboration with tiers of civil society in a good and constructive model.
3. SSCM could help to realise communities' social goals by showing a high degree of morality.
4. SSCM can help companies to obtain a competitive advantage and fortify shareholders' loyalty.

Therefore, developing countries need to reinforce sustainable activities via a suitable policy, though advanced nations use sustainability as a marketing strategy to attract ecologically and socially sensitive investors and establish an effective brand image. Although other sources of energy are accessible, industries in Egypt are responsible for sustainable SCs and must be able to meet the demands of the global community concerning cost-effectiveness, safety, and ecological conservation.

## 10. Study Limitations and Future Research Direction

This study has made a considerable contribution; it likewise has many limitations that must be recognised for other related research fields. This study had geographical limitations; hence, its current findings might not be applied widely, especially in other developing nations that are not economically comparable to Egypt. Additionally, this study was based on the participation of professionals in the Cairo and Giza construction industries. Thus, the generalizability of these results can be improved by future research, considering the extension of this study's scope to increase the coverage by involving many countries worldwide. Since it is a cross-sectional survey, this study cannot account for historical and organisational factors typical of RSSCM adoption. To comprehensively understand the association between the enablers of RSSCM adoption and the sustainable performance of projects throughout the project's lifespan, upcoming studies need to employ a longitudinal approach.

**Author Contributions:** To develop this research, E.-A.A. and M.S.U. have collaborated on the various activities. All authors have read and agreed to the published version of the manuscript.

**Funding:** The authors extend their appreciation to the Deputyship for Research & Innovation, Ministry of Education in Saudi Arabia, for funding this research work through project number (IF2/PSAU/2022/01/22491).

**Institutional Review Board Statement:** Not applicable.

**Informed Consent Statement:** Not applicable.

**Data Availability Statement:** The data derived from the study have been presented in the paper. However, further inquiries could be directed to the corresponding author.

**Conflicts of Interest:** The authors declare no conflict of interest.

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
