# Peer review of "Hybrid Assessment for Strengthening Supply Chain Resilience and Sustainability: A Comprehensive Analysis"

_sustainability, doi:10.3390/su16104010_

Round 1
Reviewer 1 Report
Comments and Suggestions for Authors
The article is devoted to the study of the issues of supply chain sustainability within the framework of sustainable development. The importance of the study is due to the currently dynamic and not always predictable changes in the economy on a national and global scale. Ensuring the sustainability of supply chains to various external factors is one of the tools for ensuring the sustainable development of the national economy. The purpose of the study is clearly formulated, the application of the research methodology is justified, the data obtained are qualitatively analyzed. The results of the study are of practical importance, the methodology proposed by the authors can be used both at the state level and at the business level to develop recommendations on ensuring the sustainability of supply chains. The material of the article is presented in good scientific language and is easily understandable. There are several recommendations that could improve the quality of the article.
1) The capacity of supply chains depends on the volume of construction work carried out by the company. It would be useful to indicate on which companies' data (in terms of construction volume) the study was carried out. This will clarify the scope of the methodology discussed in the article. Making estimates for companies of various sizes can hypothetically give different modeling results.
2) From the point of view of framing the material, you need to check: captions under the figures (Figure 2 has a caption - Figure 1), the display of formulas in the text - alignment, the use of bold text - links to sources.
In general, the article can be recommended for publication.
Comments on the Quality of English LanguageMinor editing of English language required
Author Response
Please find our responses within the attached file.

Reviewer 2 Report
Comments and Suggestions for Authors
The authors investigates the opportunity to evaluate the Egyptian businesses using 32 critical factors, extracted from the specialty literature review.
The significance of these enablers was measured and they were included into four categories.
Moreover, using a hybrid decision making technique and fuzzy logic, each category has received a weight and was prioritized.
The paper fits the Sustainability profile on perfect way.
The introduction provides sufficient background and includes enough references. The research methods are well described and the part dedicated to discussions is very well outlined.
Some aspects to review:
* Line 174: wrong reference for [55]
* Line 204: wrong reference for [61]
* Lines 237-264: wrong references for [39-46]
* Lines 183 and 195 refer to the same reference [59], but its description is different. The reference on line 195 is not the same as the bibliographic entry
* Line 291: wrong reference for [53]
* Line 391: the reference to Othman et al. (2020) research is incomplete.
* Lines 213 and 225 refer to the same reference [65], but its description is different
* Line 407: bad/missing reference
* Line 791: wrong indication of the reference. It looks like there are three authors, but instead the last one is the title of the journal
In conclusion, the paper is well constructed, but it can be published after reviewing the cited references.
Author Response

(The authors gave the same response as above.)

Reviewer 3 Report
Comments and Suggestions for Authors
In this manuscript, the authors aim to conduct an in-depth examination of the challenges hindering the adoption of sustainable and resilient supply chain management, particularly within the context of construction projects in developing countries. However, I must point out that the current draft falls short of the standards required for publication in the ‘Sustainability Journal’. My rationale for this assessment is outlined as follows:
-
Abstract Revision Required: The abstract should succinctly encapsulate the main themes and findings of the paper. It currently lacks a clear reflection of the critical elements discussed throughout the manuscript.
-
Introduction Enhancement Needed: The introduction should serve three primary purposes: highlight the research domain through current and relevant literature, identify gaps in the existing body of knowledge, and clearly present the study's objectives, findings, and structure. While there is an attempt to address the research area, the reliance on outdated sources weakens the argument. The presentation of the research gap is not persuasive, and the objectives and contributions of the paper need clearer articulation to guide the reader effectively. As it stands, the paper does not fulfill its objectives nor makes a significant contribution to the field.
-
Clarification on Sustainable Supply Chain Management (SSCM): The discussion in Section 4.1 mistakenly addresses supply chain (SC) concepts when it should focus on SSCM. It is crucial to differentiate between SC — the network of goods, services, information, and financial flows — and SCM, which optimises the entire production process for improved quality, delivery, customer satisfaction, and profitability. The literature review requires a more theoretical discourse and a critical examination of current literature using a broader array of recent sources.
-
Methodological Rigor: The methodology section is generally adequate but lacks depth in detailing the research phases. It requires a stronger academic foundation and a more comprehensive explanation of the methodological choices to ensure they align with the study’s aims and objectives.
-
Refinement of Findings and Discussion: While the findings offer interesting insights, the connection between the categorisations of enablers and the literature review claims could be more coherent. There's room for improvement in how the discussion is structured, ideally by addressing the research in relation to the question posed, acknowledging any study limitations, comparing findings with existing studies, and concluding with the implications and recommendations for future research. The utilization of recent literature is again emphasized here for engaging with contemporary discourse in the field.
-
Articulation of Contributions and Managerial Implications: It is strongly recommended that the authors explicitly state how the research contributes to existing knowledge and outline the practical implications for managers and practitioners.
These suggestions are provided with the hope of strengthening the manuscript for potential future submission. I encourage the authors to persevere with this line of inquiry, as it addresses an important area within the realm of sustainable development.
Best regards,
Author Response

(The authors gave the same response as above.)

Reviewer 4 Report
Comments and Suggestions for Authors
This study identifies the essential factors enabling resilient and sustainable supply chain management in construction projects, and uses a novel hybrid measurement of alternatives and ranks according to the compromise solution-fuzzy synthetic evaluation method. The topic is interesting. The study has developed some meaningful conclusions based on the analysis. However, I believe that there are some rooms for improvement. My main comments include:
1. The word "evaluation" is more appropriate than "decision" in the title.
2.“Background to the Research” should be changed to “Literature Review”.
3. Page 9, Lines 397-399, "Consistently, one hunted survey forms were dispersed, and six forms were filled." "Based on the rate of return principle, this accounted for 60%, dead equal for further analysis [131, 132]." The investigation process needs to be described in detail.
4. In Results, how are the 32 critical enablers of RSSCM divided into four distinct categories? The classification criteria and process should be explained.
5. The theoretical significance of this study needs to be strengthened. Emphasis is placed on the theoretical contribution of the research results.
6. The format of references needs to be corrected, as many references have formatting issues, such as [13], [14], [15], [20], [27], [29], [31], [119], [124], etc.
Comments on the Quality of English LanguageModerate editing of English language is required.
Author Response

(The authors gave the same response as above.)

Round 2
Reviewer 4 Report
Comments and Suggestions for Authors
I think the authors have made revisions to address all the issues and questions raised in the previous review. The manuscript has been improved, and it can be published.
Comments on the Quality of English LanguageMinor editing of English language required.
Author Response
Please find our response to your comments in the attached file.
